# The Effect of Curcumin on Reducing Atherogenic Risks in Obese Patients with Type 2 Diabetes: A Randomized Controlled Trial

**DOI:** 10.3390/nu16152441

**Published:** 2024-07-26

**Authors:** Metha Yaikwawong, Laddawan Jansarikit, Siwanon Jirawatnotai, Somlak Chuengsamarn

**Affiliations:** 1Department of Pharmacology, Faculty of Medicine Siriraj Hospital, Mahidol University, Bangkok 10700, Thailand; metha.yai@mahidol.ac.th (M.Y.); laddawan.jas@mahidol.ac.th (L.J.); siwanon.jir@mahidol.ac.th (S.J.); 2Siriraj Center of Research Excellence for Precision Medicine and Systems Pharmacology, Faculty of Medicine Siriraj Hospital, Mahidol University, Bangkok 10700, Thailand; 3Faculty of Pharmacy, Silpakorn University, Nakhon Prathom 73000, Thailand; 4Division of Endocrinology and Metabolism, Department of Medicine, Faculty of Medicine, HRH Princess Maha Chakri Sirindhorn Medical Center, Srinakharinwirot University, Nakhon Nayok 26120, Thailand

**Keywords:** type 2 diabetes, curcumin, atherogenic risk, cardiometabolic risks, obesity

## Abstract

Curcumin, derived from turmeric root, exhibits notable anti-inflammatory effects. These anti-inflammatory properties might also provide advantages in reducing cardiovascular complications, such as atherosclerosis. This study aimed to evaluate the efficacy of curcumin in reducing the risk of atherogenesis in obese patients with type 2 diabetes. The study employed a randomized, double-blind, placebo-controlled trial design with 227 participants diagnosed with type 2 diabetes. The parameters used to assess atherogenic risk reduction included pulse wave velocity and metabolic profiles, including low-density lipoprotein cholesterol and small dense low-density lipoprotein cholesterol. Measurements were recorded at baseline and at 3-, 6-, 9-, and 12-month intervals. After 12 months, participants receiving curcumin exhibited a significant reduction in pulse wave velocity (*p* < 0.001). This group showed significantly reduced levels of cardiometabolic risk biomarkers, including low-density lipoprotein cholesterol and small dense low-density lipoprotein cholesterol, all with *p* values less than 0.001. High-sensitivity C-reactive protein, interleukin-1 beta, interleukin-6, and tumor necrosis factor-alpha were also significantly lower in the curcumin group, with *p* values less than 0.001. The curcumin intervention significantly reduced pulse wave velocity and improved cardiometabolic risk profiles. These findings suggest that curcumin treatment may effectively reduce atherogenic risks in type 2 diabetes patients with obesity.

## 1. Introduction

Type 2 diabetes mellitus (T2DM) is a persistent state of hyperglycemia and glucose intolerance caused by an inadequate response to insulin, followed by increased insulin production and subsequent insulin resistance. With significant impacts on global public health, T2DM ranks as one of the most common chronic diseases globally, with prevalence rates soaring, particularly among adults, due to lifestyle factors. The increasing prevalence is attributed to aging populations, urbanization, sedentary lifestyles, and poor dietary habits [1,2]. Beyond its primary association with insulin resistance, T2DM encompasses abnormal metabolic conditions, including abdominal obesity, dyslipidemia, hyperuricemia, hypertension, and cardiovascular complications. 

Individuals with type 2 diabetes mellitus (T2DM) have a two- to four-times higher risk of developing cardiovascular diseases compared to those without diabetes [3]. Additionally, the prevalence of coronary heart disease in people with type 2 diabetes is about 21%, compared to 9.1% in those without diabetes [4].

Recent findings indicate that T2DM and insulin resistance are linked to cardiovascular conditions and contribute to atherogenesis [5,6,7]. This association is further elucidated through the activation of inflammatory pathways in T2DM patients, highlighting the significant role of the inflammatory milieu in the development of vascular complications and establishing the connection between diabetes mellitus and cardiovascular diseases, particularly atherosclerosis [7]. The levels of circulating pro-inflammatory cytokines, such as interleukin-1 beta (IL-1β), interleukin-6 (IL-6), and tumor necrosis factor-alpha (TNF-α), which are elevated in T2DM patients, underscore the relationship between T2DM and inflammation in the pathogenesis of atherosclerosis [8,9]. Additionally, metabolic parameters that promote atherogenesis, including abdominal obesity (total body fat and visceral fat) [10,11] and high uric acid levels [12,13], are commonly found in patients with T2DM, adding layers of complexity to their risk profile.

Natural products are crucial in developing innovative treatments for metabolic diseases like diabetes, obesity, and cardiovascular disorders. Compounds such as curcumin target multiple metabolic pathways, offering the potential for comprehensive disease management with potentially fewer side effects. Their structural diversity provides a vast pool for drug discovery. These multi-target effects and historical usage underscore the importance of natural products in creating effective, holistic treatments for metabolic diseases [14].

Curcumin (*Curcuma longa* Linn.), the principal component of the spice turmeric, has recently garnered interest for its potential benefits in addressing various health conditions, notably metabolic syndrome [15]. Research has highlighted the antihypercholesterolemic properties of curcumin extract, including a reduction in cholesterol and triglyceride (TG) levels and decreased vulnerability to low-density lipoprotein cholesterol (LDL-C). The anti-atherosclerotic and protective effects of curcumin against coronary heart disease have also been recognized [16,17,18]. Additionally, curcumin has notable potential in managing metabolic diseases due to its anti-inflammatory, antioxidant, and lipid-modifying properties. It enhances insulin sensitivity and supports weight management. Curcumin also protects liver health and offers cardiovascular benefits by reducing inflammation and oxidative stress. These multifaceted effects make it a promising complementary therapy for conditions like obesity, type 2 diabetes, and cardiovascular diseases. Enhanced formulations improve its bioavailability, making curcumin an accessible option for metabolic disease management.

Inspired by these positive findings, we proposed a human trial of curcumin treatment for the prevention of arteriosclerosis.

Our previous research indicated that curcumin intervention is associated with decreases in TG, uric acid, visceral fat, and total body fat and contributes to attenuating cardiovascular risk factors in patients with T2DM [19]. Building on these findings, our current study aimed to investigate the efficacy and safety of curcumin extract as an intervention agent for reducing atherogenesis risk in obese T2DM patients. Specifically, this study focused on assessing the anti-inflammatory effects, improvements in cardiometabolic risk markers, and weight-management effects. An evidence-based, double-blind, placebo-controlled clinical trial was conducted to explore the potential of curcumin as an intervention agent for atherogenesis in T2DM patients.

## 2. Subjects and Methods

### 2.1. Study Design and Participants

This randomized, double-blind, placebo-controlled trial was conducted at HRH Princess Maha Chakri Sirindhorn Medical Center of Srinakharinwirot University in Nakhon Nayok, Thailand. Based on specific inclusion and exclusion criteria, we selected 227 patients with T2DM. Appendix A shows a CONSORT diagram of the patient flow.

Eligible subjects were ≥35 years old and diagnosed with T2DM during the previous year. All also had well-controlled glucose levels (glycated hemoglobin [HbA1c] < 6.5%), fasting plasma glucose (FPG) < 110 mg/dL), and a body mass index ≥ 23 kg/m^2^. Diabetes was diagnosed according to the American Diabetes Association guidelines of 2017 [20]. Specifically, at least one of the following criteria needed to be met: FPG ≥ 126 mg/dL, 2-h plasma glucose ≥ 200 mg/dL during an oral glucose tolerance test, HbA1c ≥ 6.5%, or random plasma glucose ≥ 200 mg/dL accompanied by classic symptoms of hyperglycemia or a hyperglycemic crisis.

The exclusion criteria included type 1 diabetes, impaired glucose tolerance, metabolic syndrome, maturity-onset diabetes of the young, and gestational diabetes. Those with hypertension or dyslipidemia received stable doses of the respective medications to manage their conditions. To ensure the changing of variable parameters such as blood glucose levels and lipid profiles due to curcumin effects, we maintained existing treatment regimens for hyperglycemia and dyslipidemia. These unchanged regimens served as controlled parameters in the study. 

The study spanned 12 months, during which we required all participants to adhere to uniform diet and exercise protocols for the first 3 months after enrollment (the prerandomization phase). Before the study commenced, we provided participants with written lifestyle recommendations. We also conducted a 20- to 30-min individual workshop emphasizing the importance of maintaining a healthy lifestyle, including medical nutrition therapy and physical activity. Nutrient intake at baseline and at 12 weeks was estimated using Computer Dietary Guidance System Software (CDGSS 3.0) and was based on a 3-day food record (2 weekdays and 1 weekend). Further dietary habits were assessed through a questionnaire administered at baseline (Appendix A). However, we observed a relatively low daily calorie intake. This low-calorie intake could be due to an underestimation of dietary fiber consumption. Since dietary fiber is not fully digested and absorbed, it contributes fewer calories than other carbohydrates. Thus, underestimating fiber intake can lead to a seemingly lower total calorie count. Additionally, higher fiber intake increases satiety, which may reduce overall food consumption and further lower the reported calorie intake [21].

Throughout the trial, fasting overnight was mandatory before blood sample collection at 0, 3, 6, 9, and 12 months. Participants with an HbA1c ≥ 7.0% or an FPG ≥ 130 mg/dLL on two consecutive tests were excluded (Appendix A). Any occurrence of uncontrolled hypertension (blood pressure ≥ 140/90 twice) or uncontrolled dyslipidemia (LDL-C ≥ 130 mg/dL twice) also led to exclusion. 

The trial received approval from the Ethics Committee of the Faculty of Medicine, Srinakharinwirot University, Bangkok, Thailand (SWUEC/FB 4/2556, 22 February 2013), and it was registered with the Thai Clinical Trials Registry (TCTR20140303003). The research was conducted in accordance with the Declaration of Helsinki. All participants provided informed consent prior to enrollment.

### 2.2. Randomization Procedures

Participants were randomly allocated to either the curcumin-treated (intervention) group or the placebo-treated (control) group. The randomization was orchestrated using a fixed scheme based on computer-generated random numbers conducted by an independent researcher. The resulting assignment details were concealed within opaque, consecutively numbered envelopes that were sequentially opened by an independent party. Participants were made aware that the trial was comparing two distinct interventions.

### 2.3. Intervention

All participants were instructed to take three capsules of either curcumin or placebo in a double-blind manner. They were administered orally, twice daily, after meals (a total of six capsules per day) for 12 months. Each curcumin capsule contained 250 mg of curcuminoids. The curcumin and placebo capsules, which were identical in appearance, were manufactured by the Government Pharmaceutical Organization of Thailand to ensure uniformity. To assess adherence, participants were required to return all capsules at follow-up visits scheduled at 3, 6, 9, and 12 months. The number of capsules consumed by each subject was meticulously recorded (refer to Appendix A).

### 2.4. Preparation of Curcuminoid Capsules

Dried turmeric rhizomes (*Curcuma longa* Linn.), cultivated in Kanchanaburi Province, Thailand, were processed into a powder. The turmeric powder was subjected to ethanol extraction and evaporated under low pressure to produce a semisolid ethanol extract containing oleoresin and curcuminoids. The oleoresin was subsequently removed to isolate the curcuminoid extract, which had a total curcuminoid content ranging from 75% to 85%. The proportionate peak ratios of curcumin to demethoxycurcumin and bisdemethoxycurcumin within the extract were verified using high-performance thin-layer chromatography. This curcuminoid extract, which was calculated to contain 250 mg of curcuminoids, was then encapsulated under Good Manufacturing Practices standards. The detailed chemical composition and fingerprints of the extracts are shown in Appendix A.

### 2.5. Study Outcomes

The study’s primary outcome focused on assessing antiatherogenic activities by measuring pulse wave velocity (PWV) in the curcumin-treated and placebo groups. High-sensitivity C-reactive protein levels were measured to evaluate systemic inflammation and predict potential cardiac events. Furthermore, changes in the levels of pro-inflammatory cytokines (IL-1β, IL-6, and TNF-α) were meticulously recorded. Cardiometabolic risk factors, including total cholesterol, TG, LDL-C, small dense low-density lipoprotein cholesterol (sdLDL-C), and uric acid, were also monitored. Total body fat and visceral fat measurements were employed to assess abdominal obesity. Monitoring for adverse effects included observing elevations in creatinine levels ≥ 1.2 mg/dL and aspartate transaminase/alanine transaminase levels ≥ 3 times the upper limit of the normal range. Additionally, any symptoms reported by patients were thoroughly documented [22].

### 2.6. Data Collection and Measurement Methods

Measurements were conducted at baseline (prior to treatment) and at 3, 6, 9, and 12 months following the onset of the intervention. Demographic data were collected at baseline; researchers administered a questionnaire detailing medical history and medication use, and they measured body weight, height, waist circumference, and vital signs. Waist circumference, an indicator of abdominal obesity, was measured midway between the inferior margin of the rib and the superior border of the iliac crest using tape on the horizontal plane [23]. Abdominal obesity, further delineated by total body fat and visceral fat, was assessed through bio-electrical impedance analysis using a body fat analyzer (Omron HBF-362; Omron Healthcare Singapore Pte Ltd., Alexandra Technopark, Singapore) to obtain detailed analyses of body fat and visceral fat levels [24].

Blood samples were drawn at 8:00 AM from the antecubital vein of patients in a recumbent position following an overnight fast. Plasma samples for IL-1β, IL-6, and TNF-α assays were frozen and stored at −70 °C until analysis. Changes in cardiometabolic risk factors, including FPG, HbA1c, and uric acid, were measured using standard procedures over the one-year follow-up period. The homeostatic model assessment of insulin resistance (HOMA-IR) was calculated to evaluate changes in insulin resistance [25]. Plasma levels of LDL-C, sdLDL-C, high-density lipoprotein cholesterol (HDL-C), and TG were measured using diagnostic kits from Randox Laboratories Ltd. (Antrim, UK) and analyzed with an automated analyzer (spACE; Schiapparelli Biosystems Inc., Columbia, MD, USA). Apolipoprotein B levels were determined using a commercial immunoassay [26].

High-sensitivity C-reactive protein levels were quantified using a BNII Nephelometer Analyzer (Dade Behring Inc., Newark, DE, USA) with a latex-enhanced immunonephelometric assay, with a detection limit of 0.17 mg/L. Pro-inflammatory cytokine levels (IL-1β, IL-6, and TNF-α) were measured using a standard enzyme-linked immunosorbent assay according to the manufacturer’s protocol (Abcam, Cambridge, UK).

The peripheral PWV (baPWV), represented by volume waveforms for the brachium and ankle, was measured using an automated waveform analyzer (Colin VP-1000; Omron Healthcare, Kyoto, Japan), as previously described [27]. In brief, the analyzer recorded the electrocardiogram, phonocardiogram, and three pulse waves from the brachial and dorsalis pedis arteries. Signal detection was enhanced through the use of amplifier, filter, and isolation circuits. The intersecting tangent algorithm, which employs the least square mean method, was used to determine the upstroke points. Regional PWV values for the brachial and dorsalis pedis arteries were automatically calculated after collecting 10 s of data. For baPWV, the brachial–dorsalis pedis transit time was calculated by dividing the brachial–dorsalis pedis path length by the transit time. The path length was estimated as the linear distance from the sternal notch to the dorsalis pedis artery at the point of applanation.

### 2.7. Sample Size

The sample size for this study was estimated based on data from Chuengsamarn et al. [28] using a standard deviation of 160 to achieve a statistical power of 80% with an alpha error of 0.05. To show an effect on PWV levels, specifically a reduction of 60 cm/s, we determined that at least 113 subjects were needed in each treatment group to detect significant differences. After allowing for a potential 5% loss before follow-up, 227 subjects were deemed necessary across both groups. Equal group sizes were used to maximize the statistical power.

### 2.8. Statistical Analysis

Continuous data are presented as the means ± standard errors of the means, with a *p* value of <0.05 considered to indicate statistical significance. Two-tailed Student’s *t* tests were used for baseline comparisons and outcome evaluations between the two groups. Two-sided significance tests were consistently applied. The means ± standard errors of the means at 3, 6, 9, and 12 months are presented for both groups. All analyses were conducted on an intention-to-treat basis, assessing statistically significant differences between the means of the two groups at each time point. All comparisons were performed using paired samples *t*-test (for normally distributed data) or Wilcoxon signed-ranks test (for non-normally distributed data).

Categorical variables are expressed as percentages and were analyzed using the chi-square test. Outcome data on efficacy and safety included all randomized patients; they were analyzed according to their originally assigned treatment group and irrespective of the actual treatment received. Patients who lacked baseline information from their initial visit were excluded from the analysis. All the statistical analyses were performed using R software (version 4.1.2; R Foundation for Statistical Computing, Vienna, Austria).

## 3. Results

The flow of the trial is depicted in the CONSORT diagram in Appendix A. A total of 227 subjects were initially enrolled and randomly allocated to either the intervention group or the control group. There were no statistically significant differences in any baseline parameters between the curcumin-treated and placebo-treated groups (Table 1).

### 3.1. Curcumin Treatment and PWV

The mean values of the PWV for both the right and left sides were significantly lower in the curcumin-treated group than in the placebo group at the 3-, 6-, 9-, and 12-month visits (Table 2). Figure 1A,B illustrates the baseline differences in these variables between the two groups.

### 3.2. Glycemic Control Outcomes

The means of diabetes-related blood chemistries, such as HbA1c and FPG, were significantly lower in the curcumin-treated group than in the placebo group at the 6-, 9-, and 12-month visits (Table 2). The differences from baseline for these variables between the two groups are depicted in Figure 1C,D.

### 3.3. Anthropometric Measurement Outcomes

The means of waist circumference, total body fat, and visceral fat were significantly lower in the curcumin-treated group than in the placebo group at the 6-, 9-, and 12-month visits (Table 2). The differences from baseline for these measurements between the two groups are shown in Appendix A.

### 3.4. Cardiometabolic Risk Outcomes

The analysis of lipid profiles revealed that the mean levels of LDL-C and sdLDL-C, both potent markers of inflammatory processes linked to cardiovascular disease, were significantly lower in the curcumin-treated group than in the placebo group across all visits at 3, 6, 9, and 12 months (Table 2, Appendix A). Similarly, the TG/HDL-C ratio, a recognized marker for assessing cardiovascular health and risk, was significantly lower in the curcumin-treated group during these periods (Table 2, Appendix A). Additionally, apolipoprotein B, which is associated with cardiovascular disease risk, was significantly reduced in the curcumin-treated group at the 6-, 9-, and 12-month visits (Table 2, Appendix A). Serum uric acid levels were also significantly lower in the curcumin-treated group than in the control group at all study visits (3, 6, 9, and 12 months; Table 2, Appendix A).

### 3.5. Insulin Resistance and Inflammatory Biomarker Outcomes

Insulin resistance, measured via HOMA-IR, was significantly lower in the curcumin-treated group than in the placebo group at all follow-up assessments (3, 6, 9, and 12 months; Appendix A). The levels of high-sensitivity C-reactive protein, an acute-phase protein indicative of systemic inflammation, significantly decreased across all follow-up visits in the curcumin-treated group (Table 2, Appendix A). Additionally, the levels of the pro-inflammatory cytokines IL-1β, IL-6, and TNF-α were significantly lower in the curcumin-treated group than in the placebo group at the 6-, 9-, and 12-month assessments (Table 2, Appendix A).

## 4. Discussion

Cardiovascular disease remains the leading cause of morbidity and mortality globally and is largely driven by atherosclerosis. This progressive disease is characterized by the accumulation of lipids, inflammatory cells, and fibrous elements in arterial walls, leading to narrowing and compromised blood flow [29]. T2DM is intricately linked with atherosclerosis and exacerbates this condition through mechanisms such as hyperglycemia, inflammation, and lipid abnormalities. These factors damage vascular walls, foster plaque buildup, and disrupt metabolic profiles, further accelerating the progression of atherosclerosis.

Curcumin is derived from turmeric root and is known for its cardiovascular protective effects [16]. Medical doctors prescribed curcumin or placebo capsules for three-month periods. We therefore investigated its potential as a safe and well-tolerated intervention to prevent atherosclerosis in T2DM patients with obesity in Thai people. According to the World Health Organization (WHO), obesity is defined as a body mass index (BMI) of 30.0 kg/m^2^ or higher for White people. For Asian people, obesity is defined as a BMI of 25.0 kg/m^2^ or higher [30,31]. This double-blind, placebo-controlled trial evaluated ethanol-extracted curcumin, an accessible nutraceutical, with subjects consuming 1500 mg/day. We employed PWV as a primary noninvasive metric to assess arterial stiffness and, by extension, atherosclerosis risk [32,33,34]. PWV is also recognized as a reliable surrogate marker for atherosclerosis and for monitoring the efficacy of atherogenic treatments [35]. Our findings indicate that curcumin significantly reduced the PWV, substantiating its role in mitigating arterial stiffness and potential cardiovascular disease risk [36].

Furthermore, this study explored various cardiometabolic risk parameters predisposing individuals to atherosclerosis. These comprised lipid profiles (LDL-C, sdLDL-C, apolipoprotein B, TG/HDL-C ratio), abdominal obesity (waist circumference, total body fat, and visceral fat), uric acid levels, and HOMA-IR. HOMA-IR is a clinical marker of insulin resistance that strongly correlates with atherosclerosis in patients with T2DM and metabolic syndrome [37,38]. Our previous research demonstrated significant improvements in these cardiometabolic risk parameters following curcumin treatment [19].

Curcumin treatment has demonstrated anti-inflammatory effects both in vitro and in vivo [39,40,41]. It decreases the levels of IL-1β [42,43], IL-6 [44,45], and TNF-α [46,47]. These pro-inflammatory cytokines play crucial roles in the pathogenesis of atherosclerosis by promoting inflammation, endothelial dysfunction, and plaque formation. Lowering these cytokine levels represents a potential therapeutic strategy for mitigating atherosclerotic cardiovascular diseases [48,49]. Our study revealed that a 6-month curcumin treatment significantly reduced IL-1β, IL-6, and TNF-α in type 2 diabetes patients with obesity.

Additionally, we found that high-sensitivity C-reactive protein, a marker associated with inflammation and cardiovascular risk [50,51,52], was reduced after 3 months of curcumin treatment. This reduction, along with improvements in other cardiometabolic risk factors (HOMA-IR, LDL-C, sdLDL-C, apolipoprotein B, TG/HDL-C ratio, uric acid, waist circumference, total body fat, and visceral fat), suggests a comprehensive cardiometabolic benefit. These changes may contribute to the reduction in PWV observed in this study by attenuating the inflammatory processes related to atherogenesis.

In terms of diabetes management, notable decreases in FPG and HbA1c were observed as early as 3 months into treatment, suggesting an antidiabetic effect of curcumin. Regarding safety, curcumin was well tolerated at the administered dose of 1500 mg/day, consistent with other studies showing safe consumption of much higher doses (up to 8000 mg/day) without any severe side effects [53]. However, our study has several limitations. Firstly, its single-dose design prevents us from analyzing potential dose–response relationships. Secondly, being a single-center randomized controlled trial, the findings may not be generalizable to other populations or settings. 

## 5. Conclusions

Curcumin, a bioactive compound in turmeric, shows potential for reducing atherogenic risk by mitigating inflammation and improving cardiometabolic factors. These effects are promising for protecting against atherosclerosis and related cardiometabolic risks in type 2 diabetes patients with obesity. By reducing cardiometabolic risk factors, curcumin provides various health benefits, making it a valuable component of a balanced diet and healthy lifestyle. Future research should focus on optimizing its bioavailability for enhanced efficacy. We aim to broaden the findings through a multicenter randomized controlled trial.

## Figures and Tables

**Figure 1 nutrients-16-02441-f001:**
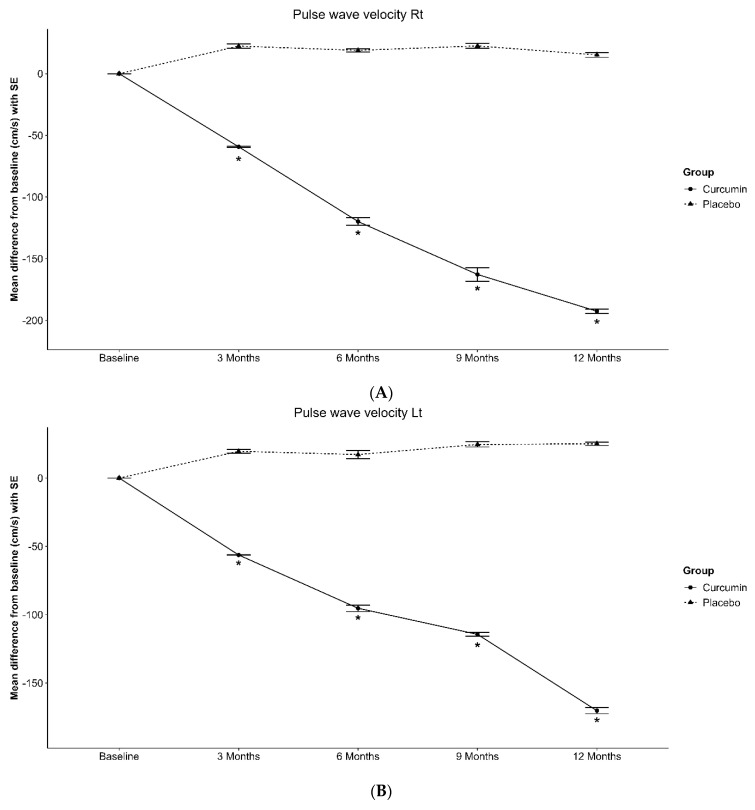
Mean of parameters with SEM at baseline, 3, 6, 9 and 12 months were compared between placebo- and curcumin-treated group. (**A**) Pulse wave velocity right; (**B**) Pulse wave velocity left; (**C**) Glycated hemoglobin (HbA 1c); (**D**) Fasting plasma glucose. * Statistically significant.

**Table 1 nutrients-16-02441-t001:** Baseline Characteristics of Study Participants.

Variables	Placebo	Curcumin	*p* *
	Mean (S.E.) (*n* = 114)	Mean (S.E.) (*n* = 113)	
Sex ratio (Males:Females)	54/80 (0.67)	62/73 (0.85)	0.87 ^ⵜ^
Age (years)	62.26 (0.81)	60.27(0.83)	0.13
BMI (kg/m^2^)	26.76 (0.38)	27.21 (0.37)	0.41
Systolic blood pressure (mmHg)	129.25(1.28)	129.76 (1.30)	0.95
Diastolic blood pressure (mmHg)	75.84 (1.05)	75.14 (1.15)	0.95
Pulse wave velocity Rt (cm/s)	1758.38 (35.26)	1812.84 (29.27)	0.26
Pulse wave velocity Lt (cm/s)	1746.13 (36.10)	1742.66 (28.40)	0.08
Total body fat (%)	33.21 (0.52)	32.66 (0.59)	0.58
Visceral fat (%)	13.27 (0.49)	13.89 (0.52)	0.30
Waist circumference (cm)	93.61(0.96)	91.97 (10.87)	0.30
FPG (mg/dl)	125.80 (2.22)	123.65 (1.73)	0.40
HbA1c (%)	6.26 (0.06)	6.28 (0.07)	0.69
HOMA-IR	5.24 (0.24)	5.38 (0.23)	0.72
TG/HDL-C ratio	2.59 (0.15)	2.53(0.16)	0.25
LDL-C (mg/dL)	103.91 (2.82)	103.10 (3.04)	0.60
sdLDL-C (mg/dL)	36.54 (1.66)	36.70 (1.64)	0.55
ApoB (mg/dL)	85.33 (1.67)	84.06 (1.23)	0.91
Uric acid (mg/dL)	5.89 (0.12)	6.07 (0.12)	0.11
hs-CRP (mg/LL)	2.98 (0.52)	3.41(0.21)	0.58
Interleukin-1 beta (pg/mL)	0.44 (0.02)	0.42 (0.02)	0.46
Interleukin-6 (pg/mL)	8.71 (0.11)	8.96 (0.12)	0.34
TNF-α (pg/mL)	5.01 (0.14)	4.78 (0.13)	0.24
Creatinine (mg/dL)	0.87 (0.02)	0.86 (0.02)	0.77
AST (U/L)	25.01 (0.87)	25.34 (0.80)	0.58
ALT (U/L)	27.58 (1.56)	30.09 (1.50)	0.08
History of cerebrovascular disease	7 (5.2%)	5 (3.7%)	0.30 ^ⵜ^
History of coronary artery disease	9 (6.7%)	8 (5.9%)	0.80 ^ⵜ^
History of hypertension	82 (61.2%)	76 (51.2%)	0.68 ^ⵜ^
History of dyslipidemia	104 (77.6%)	101 (74.8%)	0.84 ^ⵜ^

Abbreviations: FPG, fasting plasma glucose; ApoB, apolipoprotein B; ALT, alanine transaminase; AST, aspartate aminotransferase; BMI, body mass index; HbA1c, glycated hemoglobin; HDL-C, high-density lipoprotein cholesterol; HOMA-IR, homeostatic model assessment of insulin resistance; hs-CRP, high-sensitivity C-reactive protein; LDL-C, low-density lipoprotein cholesterol; Lt, left; Rt, right; sdLDL-C, small dense low-density lipoprotein cholesterol; TG, triglyceride; TNF-α, tumor necrosis factor-alpha. * Data were evaluated using the Mann–Whitney U test, except for sex (M:F ratio). ^ⵜ^ Chi-square test.

**Table 2 nutrients-16-02441-t002:** Cardiometabolic and Inflammatory Outcomes Over a 12-Month Follow-Up Period for the Placebo- and Curcumin-Treated Groups.

Outcomes	Follow-Up Period (Months)	Placebo		Curcumin		*p*
		Mean	Minimum–Maximum	Mean	Minimum–Maximum	
PWV Rt (cm/s)	0	1758.38	1074–3121	1812.84	1123–2803	NS
	3	1780.35	728–2915	1753.57	1082–3101	<0.05
	6	1777.31	950–3012	1692.96	1145–2806	<0.05
	9	1780.39	1122–2819	1650.04	1097–2527	<0.01
	12	1773.59	1089–3128	1620.20	789–2966	<0.001
PWV Lt (cm/s)	0	1746.13	760–3043	1742.66	1240–2768	NS
	3	1765.70	1064–3204	1686.39	1066–3027	<0.05
	6	1763.35	1079–3114	1647.39	1092–3010	<0.05
	9	1770.71	1195–3131	1628.27	1134–2542	<0.01
	12	1783.17	1021–2917	1572.40	1023–3055	<0.001
HbA1c (%)	0	6.26	4.80–8.90	6.28	4.40–9.50	NS
	3	6.44	5.00–8.90	6.26	4.70–9.20	<0.01
	6	6.46	5.10–9.00	6.25	4.50–8.30	<0.01
	9	6.47	5.00–10.40	6.19	4.10–8.20	<0.05
	12	6.47	5.00–10.50	6.12	4.20–8.40	<0.05
FPG (mg/dL)	0	125.08	91–285	123.65	79–178	NS
	3	128.93	100–195	124.40	80–171	NS
	6	130.34	77–231	122.82	79–204	<0.01
	9	130.93	97–201	118.67	75–165	<0.01
	12	130.71	98–194	115.49	70–160	<0.05
Waist circumference (cm)	0	93.61	61–128	91.97	71–120	<0.05
	3	93.87	62–128	91.0.1	70–118	<0.001
	6	93.99	63–165	90.44	69–135	<0.001
	9	94.81	63–129	89.39	68–117	<0.001
	12	95.46	63–140	88.70	68–114	<0.001
Total body fat (%)	0	33.21	14.40–46.10	32.66	14.40–45.30	NS
	3	33.50	16.80–46.20	32.45	14.20–45.80	<0.05
	6	34.10	16.20–47.10	31.41	13.80–43.20	<0.01
	9	34.54	19.00–47.40	31.34	13.60–46.40	<0.001
	12	35.04	21.00–48.10	31.02	13.40–46.10	<0.001
Visceral fat (%)	0	13.27	2.00–16.75	13.89	4.00–30.00	NS
	3	13.21	3.00–31.00	13.33	3.00–29.00	<0.05
	6	13.66	3.00–33.00	12.28	2.00–28.00	<0.01
	9	13.79	3.00–35.00	11.90	2.00–28.00	<0.01
	12	13.72	3.00–36.00	11.40	3.00–30.00	<0.01
LDL-C (mg/dL)	0	102.64	53–210	103.10	43.00–224.00	NS
	3	103.55	49–201	92.55	47.00–183.00	<0.01
	6	104.99	43–215	90.96	53.00–235.00	<0.001
	9	105.22	37–179	87.28	39.00–224.00	<0.001
	12	105.98	45–193	86.77	41.00–169.00	<0.001
sdLDL-C (mg/dL)	0	36.54	7.28–100.00	36.70	8.34–96.52	NS
	3	39.52	7.07–99.02	33.14	5.69–100.00	<0.01
	6	39.81	11.10–99.02	30.14	7.48–84.69	<0.01
	9	39.96	3.29–87.05	27.11	2.62–60.59	<0.001
	12	39.98	5.68–85.22	21.65	2.62–63.65	<0.001
TG/HDL-C ratio	0	2.59	0.09–2.98	2.53	0.20–1.99	NS
	3	3.20	0.29–2.39	2.28	0.16–2.54	<0.01
	6	3.53	0.26–2.46	2.26	0.20–2.51	<0.001
	9	3.67	0.33–2.30	2.18	0.18–2.55	<0.001
	12	3.72	0.27–2.51	2.12	0.17–2.54	<0.001
ApoB (mg/dL)	0	84.33	35–153	84.06	35.00–120.00	NS
	3	84.30	36–153	78.54	35–132	<0.05
	6 months	84.67	34–150	60.68	29–105	<0.001
	9 months	84.66	33–150	51.51	21–85	<0.001
	12 months	84.66	38–153	41.35	20–78	<0.001
Uric acid (mg/dL)	0 months	5.82	3.03–10.75	5.99	2.31–10.25	NS
	3 months	6.02	2.74–9.41	5.87	2.72–11.13	<0.05
	6 months	6.58	3.76–11.43	5.95	2.59–9.99	<0.01
	9 months	6.54	3.40–10.61	5.47	2.71–9.21	<0.001
	12 months	6.56	3.23–10.78	5.05	2.32–8.76	<0.001
HOMA-IR	0 months	5.24	1.70–21.80	5.38	1.20–14.20	NS
	3 months	5.88	2–17	5.25	1.70–12.80	<0.05
	6 months	5.93	1.80–17.90	5.17	1.60–16.50	<0.05
	9 months	6.02	2.20–19.80	5.02	1.30–11.50	<0.05
	12 months	6.04	2.30–18.00	4.86	1.20–11.00	<0.05
hs-CRP (mg/L)	0 months	3.41	0.20–44.49	2.98	0.22–16.64	NS
	3 months	3.51	0.17–14.10	2.94	0.29–12.52	<0.05
	6 months	3.53	0.26–29.81	2.81	0.26–19.25	<0.05
	9 months	3.68	0.31–36.64	2.69	0.19–44.75	<0.001
	12 months	3.75	0.25–30.24	2.60	0.14–14.60	<0.001
Interleukin-1β (pg/mL)	0 months	0.44	0.01–0.86	0.46	0.01–0.88	NS
	3 months	0.46	0.02–0.87	0.45	0.01–0.87	NS
	6 months	0.71	0.20–1.74	0.43	0.15–1.54	<0.001
	9 months	0.72	0.20–1.65	0.41	0.12–0.99	<0.001
	12 months	074	0.32–1.86	0.31	0.10–1.39	<0.001
Interleukin-6 (pg/mL)	0 months	8.71	7.04–10.56	8.96	7.04–10.56	NS
	3 months	8.89	7.04–10.56	8.72	7.04–10.56	NS
	6 months	12.84	5.21–17.99	7.54	3.11–14.99	<0.001
	9 months	14.30	7.65–19.66	6.82	3.2–13.24	<0.001
	12 months	15.84	4.33–19.66	6.12	3.09–12.40	<0.001
TNF-α (pg/mL)	0 months	5.01	2.64–7.04	4.77	2.64–7.04	NS
	3 months	5.16	2.64–7.04	4.84	2.64–7.04	NS
	6 months	5.91	2.18–14.88	4.23	1.46–10.5	<0.001
	9 months	6.37	2.24–14.98	3.81	1.43–9.44	<0.001
	12 months	6.77	2.14–15.37	3.46	1.33–8.59	<0.001

Abbreviations: FPG, fasting plasma glucose; ApoB, apolipoprotein B; HbA1c, glycated hemoglobin; HDL-C, high-density lipoprotein cholesterol; HOMA-IR, homeostatic model assessment of insulin resistance; hs-CRP, high-sensitivity C-reactive protein; LDL-C, low-density lipoprotein cholesterol; Lt, left; NS, not statistically significant; PWV, pulse wave velocity; Rt, right; sdLDL-C, small dense low-density lipoprotein cholesterol; TG, triglyceride; TNF-α, tumor necrosis factor-alpha.

## Data Availability

The original contributions presented in the study are included in the article/Appendix A, further inquiries can be directed to the corresponding author.

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
