# Peer review of "The Effect of Curcumin on Reducing Atherogenic Risks in Obese Patients with Type 2 Diabetes: A Randomized Controlled Trial"

_nutrients, 2024, doi:10.3390/nu16152441_

Round 1
Reviewer 1 Report
Comments and Suggestions for Authors
Type of manuscript: Article
Title: Curcumin Extract Diminishes Atherogenic Risk in Type 2 Diabetes Mellitus Patients With Obesity
The subject of this paper is interesting. It is well written, and the design used is appropriate. The results obtained are worthy of note.
- According to participant’s BMI, they present overweight, not obesity. This should be corrected along the manuscript.
- Their intake of dietary is very low. I would suggest authors to explain why could this be. Could it be that dietary fibre intake was underestimated?
- It would be better to explain with more detail how the volunteers consumed the capsules. What time were they taken? Where the capsules consumed separated from meals?
- The sample size paragraph needs to be completed. Which was the main variable considered? And the difference between used to estimate the sample size? What statistical power was used? This point is very important, and it has not been addressed adequately.
- The quality of the chromatogram is poor. It should be replaced with a better photo.
- The limitations of the study have not been included in the manuscript.
- I miss a sentence at the end stating that the supplements should be used in the context of a healthy diet and lifestyle.
Author Response
Reviewer 1
Comment 1:
According to participant’s BMI, they present overweight, not obesity. This should be corrected along the manuscript.
Response:
Thank you very much for the suggestions. According to WHO criteria for Asian, we classified the type 2 diabetes patients were obesity. To specify this, we added the sentences in the revised manuscript as highlighted (line 366-371).
“We therefore investigated its potential as a safe and well-tolerated intervention to prevent atherosclerosis in T2DM patients with obesity in Thai people. According to the World Health Organization (WHO), obesity is defined as a body mass index (BMI) of 30.0 kg/m² or higher for White people. For Asian people, obesity is defined as a BMI of 25.0 kg/m² or higher 1, 2.
Comment 2:
Their intake of dietary is very low. I would suggest authors to explain why could this be. Could it be that dietary fibre intake was underestimated?
Response:
Thank you very much for the valuable suggestion. To respond to the reviewer’s suggestion, we discussed the reduction of BMI in the treatment group as highlighted (line 127-133).
“However, we observed a relatively low daily calorie intake. This low-calorie intake could be due to an underestimation of dietary fiber consumption. Since dietary fiber is not fully digested and absorbed, it contributes fewer calories than other carbohydrates. Thus, underestimating fiber intake can lead to a seemingly lower total calorie count. Additionally, higher fiber intake increases satiety, which may reduce overall food consumption and further lower the reported calorie intake 3.”
Comment 3:
It would be better to explain with more detail how the volunteers consumed the capsules. What time were they taken? Where the capsules consumed separated from meals?
Response:
Thank you very much for the suggestion. We added the information as highlighted (line 154-155).
“They were administered orally, twice daily, after meals (a total of six capsules per day) for 12 months.”
Comment 4:
The sample size paragraph needs to be completed. Which was the main variable considered? And the difference between used to estimate the sample size? What statistical power was used? This point is very important, and it has not been addressed adequately.
Response:
Thank you very much for the suggestions, we added the sentence as highlighted (line 225-228).
“Using a standard deviation of 160 to achieve a statistical power of 80% with an alpha error of 0.05. To show an effect on PWV levels, specifically a reduction of 60 cm/s, we determined that at least 113 subjects were needed in each treatment group to detect significant differences.”
Comment 5:
The quality of the chromatogram is poor. It should be replaced with a better photo.
Response:
We improved the quality of the chromatogram as shown in Supplementary Figure S2.
Comment 6:
The limitations of the study have not been included in the manuscript.
Response:
Thank you very much for the suggestions, we added the limitations of the study as highlighted (line 403-406).
“However, our study has several limitations. Firstly, its single-dose design prevents us from analyzing potential dose-response relationships. Secondly, being a single-center randomized controlled trial, the findings may not be generalizable to other populations or settings.”
Comment 7:
I miss a sentence at the end stating that the supplements should be used in the context of a healthy diet and lifestyle.
Response:
Thank you very much for the suggestion. In the revised version, we added the limitations of the study as highlighted (line 411-413).
“By reducing cardiometabolic risk factors, curcumin provides various health benefits, making it a valuable component of a balanced diet and healthy lifestyle”.

Reviewer 2 Report
Comments and Suggestions for Authors
Thank you for allowing me to evaluate this interesting and well written manuscript. The text makes a good read and clearly explains the studies.
Some suggestions:
Page 5. Sample size. Of the parameter the sample size is based on, the SD is given ('160'). Which endpoint (study parameter) was used for the sample size calculation? I assume it is the (primary endpoint) PWV(?). This must be mentioned.
In the discussion, it may be of clinical interest to dedicate a few lines of text on how the curcumin is to be distributed to patients. Will this be on prescription by medical dr's or will it be provided over the counter?
The tables are clear and abbreviations are well described. Very informative and clear figures both in the main manuscript texts and the suppl. data sheet.
I liked the many graphical displays; however for publishing there may be a length issue, mainly due to the many figures. If too lengthy, consider to shift some of the figures to the suppl. material.
Otherwise, I have no comments or suggestions on this interesting and well written paper.
Comments on the Quality of English Language
None
Author Response
Reviewer 2
Comment 1:
Page 5. Sample size. Of the parameter the sample size is based on, the SD is given ('160'). Which endpoint (study parameter) was used for the sample size calculation? I assume it is the (primary endpoint) PWV(?). This must be mentioned.
Response:
Thank you very much for the suggestions. We added the sentence as highlighted (line 225-228).
“Using a standard deviation of 160 to achieve a statistical power of 80% with an alpha error of 0.05. To show an effect on PWV levels, specifically a reduction of 60 cm/s, we determined that at least 113 subjects were needed in each treatment group to detect significant differences.”
Comment 2:
In the discussion, it may be of clinical interest to dedicate a few lines of text on how the curcumin is to be distributed to patients. Will this be on prescription by medical dr's or will it be provided over the counter?
Response:
Thank you very much for the comment and questions. We added the information as highlighted (line 366-367).
“Medical doctors prescribed curcumin or placebo capsules for three-month periods.”
Comment 3:
I liked the many graphical displays; however, for publishing there may be a length issue, mainly due to the many figures. If too lengthy, consider to shift some of the figures to the suppl. material.
Response:
Thank you very much for the suggestion. We reduced the number of figures presented in the manuscript to 4 (Fig1A-D). the remaining figures were in the supplementary Figure S3 as followings: -
The differences from baseline for these measurements between the two groups are shown in Supplementary Figure S3 A-C (line 333-334).
The analysis of lipid profiles revealed that the mean levels of LDL-C and sdLDL-C, both potent markers of inflammatory processes linked to cardiovascular disease, were significantly lower in the curcumin-treated group than in the placebo group across all visits at 3, 6, 9, and 12 months (Table 2, Supplementary Figure S3 D-E) (line 336-339).
Similarly, the TG/HDL-C ratio, a recognized marker for assessing cardiovascular health and risk, was significantly lower in the curcumin-treated group during these periods (Table 2, Supplementary Figure S3 F) (line 339-342).
Additionally, apolipoprotein B, which is associated with cardiovascular disease risk, was significantly reduced in the curcumin-treated group at the 6-, 9-, and 12-month visits (Table 2, Supplementary Figure S3 G) (line 342-344).
Serum uric acid levels were also significantly lower in the curcumin-treated group than in the control group at all study visits (3, 6, 9, and 12 months; Table 2, Supplementary Figure S3 H) (line 344-346).
Insulin resistance, measured via HOMA-IR, was significantly lower in the curcumin-treated group than in the placebo group at all follow-up assessments (3, 6, 9, and 12 months; Supplementary Figure S3 I) (line 348-350).
The levels of high-sensitivity C-reactive protein, an acute-phase protein indicative of systemic inflammation, significantly decreased across all follow-up visits in the curcumin-treated group (Table 2, Supplementary Figure S3 J) (line 350-352).
Additionally, the levels of the pro-inflammatory cytokines IL-1β, IL-6, and TNF-α were significantly lower in the curcumin-treated group than in the placebo group at the 6-, 9-, and 12-month assessments (Table 2, Supplementary Figure S3 K-M) (line 352-355).

Reviewer 3 Report
Comments and Suggestions for Authors
Comments to Authors
1. Title: keep the more attractive and comprehensive one.
2. Please add a graphical abstract summarizing the manuscript contents according to the journal instructions.
3. Abstract: abstract should be a total of 200 words maximum. The abstract should be a single paragraph and should follow the style without headings. Please follow the instructions for authors,
4. Introduction: please add the statistics of atherogenic risks with diabetes (prevalence and incidence) and importance of natural products for developing innovative medications and treatment methods against metabolic diseases.
5. Please add an explanation of the application of controlled parameters including lipid profile and glucose level.
Conclusion needs to be well written and please ass future prospects to it. References: please follow the style of references according to journal instruction
Comments on the Quality of English Language
English editing is highly recommended
Author Response
Reviewer 3
Comment 1:
Title: keep the more attractive and comprehensive one.
Response:
Thank you very much for the suggestions, we changed the title of manuscript to “The Effect of Curcumin on Reducing Cardiometabolic Risks in Type 2 Diabetes Mellitus Patients: A Randomized Controlled Trial” as highlighted (line 2-4).
Comment 2:
Please add a graphical abstract summarizing the manuscript contents according to the journal instructions.
Response:
In the revised version, we added graphical abstract as suggested.
Comment 3:
Abstract: abstract should be a total of 200 words maximum. The abstract should be a single paragraph and should follow the style without headings. Please follow the instructions for authors.
Response:
Thank you very much for the valuable suggestions. To respond to the reviewer’s suggestions, we corrected the abstract according to the instructions for authors and the total word count is now 199 words, as highlighted (line 19-34).
“Abstract
Curcumin, derived from turmeric root, exhibits notable anti-inflammatory effects. These anti-inflammatory properties might also provide advantages in reducing cardiovascular complications, such as atherosclerosis. This study aimed to evaluate the efficacy of curcumin in decreasing the risk of atherogenesis in obese patients with type 2 diabetes. The study employed a randomized, double-blind, placebo-controlled trial design with 227 participants diagnosed with type 2 diabetes. The parameters used to assess atherogenic risk reduction included pulse wave velocity and metabolic profiles, including low-density lipoprotein cholesterol and small, dense low-density lipoprotein cholesterol. Measurements were recorded at baseline and at 3-, 6-, 9-, and 12-month intervals. After 12 months, participants receiving curcumin exhibited a significant reduction in pulse wave velocity (P < 0.001). This group showed significantly reduced levels of cardiometabolic risk biomarkers, including low-density lipoprotein cholesterol and small, dense low-density lipoprotein cholesterol, all with P values less than 0.001. High-sensitivity C-reactive protein, interleukin-1 beta, interleukin-6, and tumor necrosis factor-alpha were also significantly lower in the curcumin group, with P values less than 0.001. The curcumin intervention significantly reduced pulse wave velocity and improved cardiometabolic risk profiles. These findings suggest that curcumin treatment may effectively reduce atherogenic risks in type 2 diabetes patients with obesity.”
Comment 4:
Introduction: please add the statistics of atherogenic risks with diabetes (prevalence and incidence) and importance of natural products for developing innovative medications and treatment methods against metabolic diseases.
Response:
Thank you very much for the suggestions. To respond to the reviewer’s suggestions, we added the sentences in the revised manuscript as highlighted (line 47-50).
“Individual with type 2 diabetes mellitus (T2DM) have a 2 to 4 times higher risk of developing cardiovascular diseases compared to those without diabetes 4. Additionally, the prevalence of coronary heart disease in people with type 2 diabetes is about 21%, compared to 9.1% in those without diabetes 5.”
Comment 5:
Please add an explanation of the application of controlled parameters including lipid profile and glucose level.
Response:
Thank you very much for the suggestion. To respond to the reviewer’s suggestions, we added the sentences in the revised manuscript for blood glucose control as highlighted (line 115-118).
“To ensure the changing of variable parameters such as blood glucose levels and lipid profiles were due to curcumin effects, we maintained existing treatment regimens for hyperglycemia and dyslipidemia. These unchanged regimens served as controlled parameters in the study.”
Comment 6:
Conclusion needs to be well written and please add future prospects to it.
Response:
Thank you very much for the suggestions. In the revised version, we adjusted the conclusion the revised manuscript as highlighted (409-416).
“5. Conclusions
Curcumin, a bioactive compound in turmeric, shows potential for reducing atherogenic risk by mitigating inflammation and improving cardiometabolic factors. These effects are promising for protecting against atherosclerosis and related cardiometabolic risks in type 2 diabetes patients with obesity. By reducing cardiometabolic risk factors, curcumin provides various health benefits, making it a valuable component of a balanced diet and healthy lifestyle. Future research should focus on optimizing its bioavailability for enhanced efficacy. We are hoping to expand the results of multicenter randomized controlled trial.”
Comment 7:
References: please follow the style of references according to journal instruction
Response:
Thank you very much for the suggestions. To respond to the reviewer’s suggestions, we adjusted the style of references according to journal instruction (423-544).
